# Learning Multiplication by Translating across Microworlds

Sheena Tan , Sean Chorney  and Nathalie Sinclair *

Faculty of Education, Simon Fraser University, Burnaby, BC V5A 1S6, Canada;
sheena_miao_ying_tan@sfu.ca (S.T.); sean_chorney@sfu.ca (S.C.)
* Correspondence: nathsinc@sfu.ca

**Abstract:** In this article, we explore students' experiences of using two different digital microworlds of multiplication, which can be found in the multitouch application TouchTimes. We draw on Diagne's notion of translation to frame our study, focusing on the learning that occurs in the movement between the two microworlds. We study translation in terms of actions, strategies, perceptions, and preferences and highlight both the translatables and the untranslatables that emerged in the pair-based interviews that were conducted with grades 3–4 students.

**Keywords:** translation; microworld; multiplication; TouchTimes; gesture

## 1. Introduction

While teachers are often encouraged to present students with multiple models or representations of multiplication (groups of, area model, number lines, etc.), some researchers have recently recommended the use of one single model or representation, purportedly to simplify both the structure and learning of multiplication. Izsák and Beckmann [1], for example, attempts to create a single unified approach to multiplication, based on measurement, which would connect the underlying structure of various types of multiplicative problems. More recently, Foster [2] argues that multiplication should be taught with one single coherent representation, that of the number line, contending that it would provide a coherent visual approach to multiplication that would work across all real numbers (not only whole numbers). Foster also suggests that introducing other representations may lead to confusion not only because there would be more representations to learn—each of which take time—but also because not all representations are mathematically correct. For example, he criticises the area model and argues that students who use that model often confound linear and area units, which can be avoided with the use of "a pair of mutually perpendicular number lines, comprising Cartesian axes" (p. 24). With the use of Cartesian graphs, a number will always be represented by the length of a line segment.

Both Izsák and Beckmann and Foster are focused on simplifying the learning of multiplication, which is presented as a fixed, abstract concept—that is, it exists independently of its representations. It is further assumed that there are universal representations that can capture all the significant structures of multiplication. We have become interested in these arguments through our own design of and research on *TouchTimes* [3], which is a multitouch application designed to support students' learning of multiplication. Crucially, it contains two distinct and dynamic microworlds, a term introduced by Papert in 1972 to designate "a model of a domain of mathematical knowledge" [4] (p. 65). We use the term microworld instead of model to point to its interactive, multimodal, open-ended, and embodied nature. Each of the two microworlds instantiates multiplication in different ways—one rendering multiplication as a change in units (with significant Davydovian influences) and the other on the functional aspect of multiplication based on the Cartesian approach. The inclusion of two microworlds in *TouchTimes* is in part premised on the assumption, following Davis and Renert [5], that multiplication is an open concept that is continually being adjusted, therefore suggesting that students should not only be aware of different ways of thinking of

multiplication, but also able to move between them. It is also premised on a socio-material conception of mathematics (as per ref. [6]) at takes mathematical concepts to be situated within specific practices, which involve a network of tools, such as paper, pencil, abaci, grids, symbols, calculators, etc. and sociocultural norms (in terms of techniques applied, artefacts produced, etc. (see also Maffia's [7], response to ref. [2])). We hypothesise that the use of multiple models or representations (or microworlds, in our case) can also better support inclusive efforts in education, offering a plurality of ways of understanding concepts such as multiplication [8].

Given the importance we attach to the value of plurality and to the dearth of research on how exactly students are assumed to coordinate across models and representations, we focus in this paper on the question of how to conceptualise the notion of coordination. To do so, we consider recent approaches within the broad umbrella of semiotic mediation, which focus on identifying the ways in which students make semiotic links. Semiotic pertains to any type of sign, including words and gestures as well as diagrams, images and artefacts. Researchers who focus on the important role of Vygotskian mediation in learning often pay specific attention to these signs, particularly when focusing on the use of different visual representations, on technologies (including paper-and-pencil diagrams as well as dynamic images) and on communication (oral language as well as bodily movements). We expand this semiotic approach to account more broadly for the embodied and affective aspects of experience that are significantly at play in our tangible environment of *TouchTimes* (TT). We do so by introducing the notion of *learning as translating*, which is a perspective that conceptualises learning as occurring just as much in the passage from one model/representation/microworld to another as in the experience of each one and is inspired by the philosopher Souleymane Bachir Diagne [9].

We begin with a short overview of the recent research on teaching and learning multiplication, particularly in relation to the design of TT. We then introduce the work that has influenced our thinking about the use of multiple models/representations, particularly in relation to the duo of artefacts approach [10] and to the concepts of semiotic synergy [11] and semiotic interference [12]. Given our interest in studying how students coordinate different models/representations/microworlds, we also draw on Nemirovsky's [13] expanded notion of transfer. Instead of focusing on how one schema or concept is acquired in one context and then used in another, Nemirovsky takes a non-acquisitionist perspective by studying how one experience folds into another, where experience involves not only cognition, but also the social, emotional, and aesthetic. We resituate Nemirovsky's transfer in terms of translation, in the sense of Diagne [9], because this approach directs attention to the back-and-forth movement between two languages (or, in our case, between two microworlds). After outlining our theoretical approach, we will describe the main functionalities of TT. We strongly recommend that readers download and experiment with TT before reading on—videos are also available at touchtimes.ca. Finally, we analyse the way in which four pairs of grades 3–4 students (from the same mixed-grade classroom) engaged in translation as they responded to a set of tasks specifically designed to encourage back-and-forth movement across the two microworlds. We conclude with some pedagogical and theoretical considerations.

## 2. On Learning Multiplication with Multiple Models and Representations

There has been a recent flurry of interest in models and representations (since it is not clear how these authors distinguish models from representations, we adopt their language in this section, but will use the term microworld when discussing our own research) of multiplication [1,14,15]. Much of this literature is focused on understanding models and representations, so that they can be used efficiently and effectively in teaching children multiplication. For example, Maffia and Mariotti explore two formal models of multiplication, the repeated-sum model and the array model, and discuss how that understanding is productive for teaching multiplication to students. In particular, they suggest that tacit qualities of each model will stress specific and distinct properties of

multiplication. Similarly, with pedagogy as central to their inquiry, Larsson [16] studied grade 6 students' conceptions of distributivity and commutativity and how different models supported or hindered understanding of those practices. Both Maffia and Mariotti and Larsson found that the equal group method (unary, according to ref. [17]) introduced challenges for commutativity activity.

In another study that focuses on visual models, Kosko [14] performed a large-scale quantitative study looking at 182 elementary students and their reasoning of multiplication when presented with set area and length visual representations. Similar to the studies mentioned above, Kosko argues that visual representations can promote or limit engagement in multiplicative reasoning and each representation will highlight some elements over others. His results suggest significant differences in the representations related to how the multiplicative unit was presented and could be varied.

In each of these studies, models and representations are implicitly compared by assessing students' practices and conceptions when using different ones. There seems to be broad agreement that any given model or representation will convey certain features of multiplication. In these studies, however, each one is analysed separately from others. There is little inquiry into how students manage the use of more than one model or representation or how they compare them, choose them, or think about them differently. Finally, there is little discussion of pedagogical questions of how teachers can support students' coordination of multiple models or representations.

According to Askew [18], no mediating artefact (be it a model or representation) is inherently multiplicative; students need to shape them multiplicatively. He argues that students who are left to make their own interpretations may rely solely on additive reasoning. Additionally, as Downton et al. [19] show, without support from adequate models/representations or teachers, students will often rely on algorithmic approaches to handling multiplication, which can lead to errors when they are solving multiplication word problems. This appears to be one of the consequences of approaching multiplication through a symbolic approach. By having students develop their own sense of multiplication through an interactive, multimodal embodied experience, we suggest that even if they create their own algorithms, they will still be more meaningful.

## 3. Theoretical Framing

Research on the use of digital technologies has typically focused on the specific affordances of the technology, often contrasting them with those found in paper-and-pencil environments. However, in the past decade, more researchers have become interested in how students might work with different environments and what learning opportunities there might be in combining both digital and non-digital resources. For example, Maschietto and Soury-Lavergne's [10] construct of the "duo of artefacts" was developed through their studies on the use of both digital and non-digital environments in the learning of specific concepts. Crucially, their environments were actually quite similar (such as the physical manipulative called the Pascaline and a digital version created in a dynamic geometry environment). They studied the different signs (such as the use of arrows to represent movement or direction) that arose while students used both environments, highlighting the different signs that were available in each environment and showing how the combination of the artefacts enabled students to engage in a more fulsome interaction with the target concept.

Drawing on the theory of semiotic mediation, Faggiano, Montone, and Mariotti [11] similarly designed tasks that encourage students to reflect on the use of one artefact (paper folding) while using another (dynamic geometry). They describe semiotic chains that are constructed that involve signs related to both artefacts, which they describe as synergy. Interestingly, unlike Machietto and Soury-Lavergne, who are attentive to the signs that differ across artefacts—particularly the material signs relating to how students move and interact with the artefacts—and thus allow for a more diffusive image of concepts, Faggiano et al.'s notion of synergy is more concerned with similarity and as such with

students developing a coherent, universal understanding of symmetry through their experiences in both environments. Similarly oriented towards coherence and universality, Panorkou, York, and Germia [20] combine the constructs of "duo of artefacts" and synergy to study students' transitions between three representations (simulation, table, and graph) involving covarying quantities. They find that synergies can be productive for students to develop and reorganise their understanding.

Drawing on Peircean semiotics, Maffia and Maracci [12] use the construct of semiotic interference to study "how the mathematical learning process triggered by the use of an artefact in the classroom can be affected by the use of another one, even in different moments and for different purposes" (p. 57). They also make use of the notion of a semiotic change to describe how "signs are continuously translated into new signs" as "students construct shared meanings that converge towards the mathematical community's interpretation" (ibid). Semiotic interference is said to occur between two artefacts "when the interpretant of a sign whose object belong to the context of an artefact is translated by a student in a new sign whose object belong to the context of another artefact" (p. 58). For example, the sign in the form of the word "tie" was first used to represent the (concrete) tying of ten straws into a group. Subsequently, the sign "tie" got translated by a student into an operation of "changing 10 unit-beads with 1 tens-bead" (p. 61) through a virtual tying of beads in the context of an abacus. For these authors then, the coordination between artefacts occurs through semiotic interference, which may need to be explicitly triggered.

Like Faggiano and colleagues, the emphasis remains on convergence, since the signs that do not become part of the semiotic chain are ignored. However, we note the attention paid to the passage between one artefact and the other. Indeed, the authors use the word "translate" in the quote above, which is precisely our phenomenon of interest. Given the connection between translation and language, it is not surprising that the majority of the semiotic chains identified in the studies by Maffia and Maracci are indeed specific words. While specific words are likely to be quite significant in students' moving back and forth between models, we are interested in a more general sense of how students coordinate their experiences with each model. This can be found in Nemirovsky's [13] broader sense of transfer, which attends not only to linguistic ways in which one experience can become a part of another, but also social, emotional, and embodied ways. This is compatible with the inclusive materialism approach of de Freitas and Sinclair [6], which stresses the socio-material nature of mathematics concepts. Rather than being strictly discursive, mathematical concepts are embodied and situated within certain practices of sense-making. This means that moving between two microworlds will involve not just certain ways of talking, but also certain ways of moving, or feeling, and of knowing. Note that this theoretical perspective does not see concepts as fixed, neither as platonic objects or through sociocultural determinations, but instead as materially constituted and therefore changing with different socio-material environments. This is particularly relevant to the context of TT, which provides a novel material environment for performing multiplication. We are therefore interested in broadening the Maffia and Maracci metaphor of translation to more fully account for the types of coordination involved in learning across artefacts. Further, rather than assuming the existence of a single concept that somehow encompasses all its different instantiations, which drives attention to the learning of each instantiation, we posit that it is precisely in the movement between instantiations, in the translation, that significant aspects of learning occur.

This perspective arises from the work of the philosopher Souleymane Bachir Diagne on translation, which is concerned not only with epistemology (how do we know each other?) but with ethics (how do we come to know each other in ways that create and maintain relations of reciprocity?). Since Diagne is concerned with questions of cultural and racial inequity and how they might be addressed through translation, his work might seem inappropriate for our context, where we are concerned with how students coordinate different meanings of multiplication as they move from one microworld-situation to another. However, Diagne's work helps illuminate some significant aspects of translation that we

think are relevant to the study of mathematical learning. First, he insists on a different kind of universality, one that brings together a plurality of languages through translation, which is different from the universality that seeks the singular, overarching meaning. With this comes a commitment to opacity, that is, that there will never be complete translation between languages (or cultures)—that there are untranslatables that must nevertheless be respected as meaningful. Additionally, translation does not just occur through language, but through visions of the world, which will include beliefs, values, and habits. Finally, that the current inequities between dominant and dominated languages necessitate increased efforts of translation.

In our context of school mathematics, the dominant language of multiplication is symbolic; in contrast, TT supports a gestural form of expression as well as a visual and dynamic one. Another aspect of the dominant language of multiplication is that of "groups of"; with TT, however, we propose multiple languages of multiplication and the means of translating between them, which is less about replacing "groups of" thinking than offering alternatives. Finally, when Diagne asserts that understanding is translating (citing George Steiner), we interpret this, in an analogical way, as saying that understanding multiplication is less about acquiring a particular meaning of multiplication or participating in a particular discourse and more about translating across ways of thinking, of communicating, of acting. In a sense, we are reversing the transfer paradox by taking transfer itself—as a process—as the locus of learning and understanding.

To study learning as translation, we will thus be conceiving the two microworlds in our study as two visions of the world (of multiplication), staying attentive to the variety of signs they mobilise, as well as to beliefs, values, and habits they modulate. There are similarities in languages that lay a foundation for communication, for ways of seeing, for ways of thinking, yet each language is different enough from other languages to earn a sense of uniqueness. We see the microworlds of TT as different languages with their own set of practices, which include not only the words and gestures that are used, but also the material context, which might include the ways of acting and moving that arise in the material environment (which include the artefacts, the space, the people, etc.). Each of these actions are part of the language of the microworld. Another way of respecting the language of the microworld is to acknowledge meanings that do not converge, but that are still part of students' experiences of multiplication.

## 4. Methodological Approach

We first describe the two microworlds (Grasplify and Zaplify), to provide readers with a better sense of the specific ways in which they instantiate multiplication. Again, readers are encouraged to download the free version of TT on their iPads and explore it for themselves in order to get a much better sense of how the application works. After describing TT, we will elaborate on the context in which this study is situated.

### 4.1. TouchTimes—Two Microworlds of Multiplication

*TouchTimes* is a multi-touch iPad application that is designed to support students' learning of multiplication. It comprises two complementary microworlds, Grasplify and Zaplify, where microworlds are immersive learning environments in which students can make decisions and explore the implications of those decisions. Papert describes a microworld as "learn[ing] to transfer habits of exploration from their personal lives to the formal domain of scientific theory construction" [21] (p. 117). The TT microworlds highlight different properties of multiplication in visually different ways, providing students the opportunity to explore two different approaches and develop their multiplicative thinking.

Grasplify's design was influenced by Davydov's [22] approach where multiplication is characterised by a double change-in-units. The first unit is the multiplicand (quantity of a unit), which is established first, followed by the multiplier (number of units) as the second unit. Specifically, in Grasplify, there are two different objects, pips and pods, that can be created by touching our fingers on either side of the screen. When one or more fingers

first touch either side of the screen, pips are created (refer to Figure 1a where 4 pips are created), which is the first instance of unitising. When one or more fingers are pressed on the other side of the screen, pods are created (refer to Figure 1b where 3 pods are created), resulting in the second unitising. Multiple pips and pods can be created by pressing multiple fingers all at once or by tapping one finger repeatedly on the respective sides. When one pip is added or removed, each pod changes accordingly to contain one more or one less pip. Consequently, performing multiplication in Grasplify involves "a count of a [larger] unit for which a relationship to another, smaller unit, is already established" (p. 12).

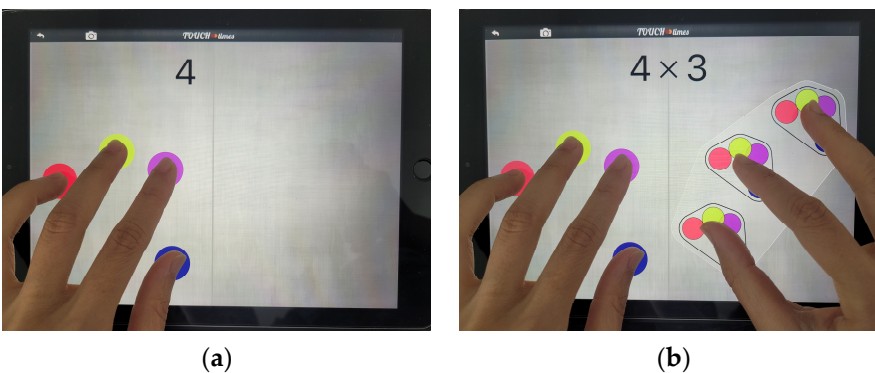

| (a) | (b) |

**Figure 1.** (**a**) Making 4 pips; (**b**) making 3 pods.

The design of Zaplify was influenced by Vergnaud's [17] work on the functional and relational aspects of multiplication, which includes a focus on doubling, tripling, etc. In Zaplify, when one or more fingers are placed on the left of the screen, horizontal lines or lightning bolts are created (refer to Figure 2a where 4 horizontal lines are created). When one or more fingers are placed at the bottom of the screen, vertical lines or lightning bolts are created (refer to Figure 2b where 3 vertical lines are created). The lines represent each factor in the multiplication, and the points of intersection between them represent the product of the multiplication. Any change to one factor will change the product as it distributes over the other factor. As a result, multiplication in Zaplify emphasises the multiplicative relationship between the factors and the product while portraying an array model of multiplication.

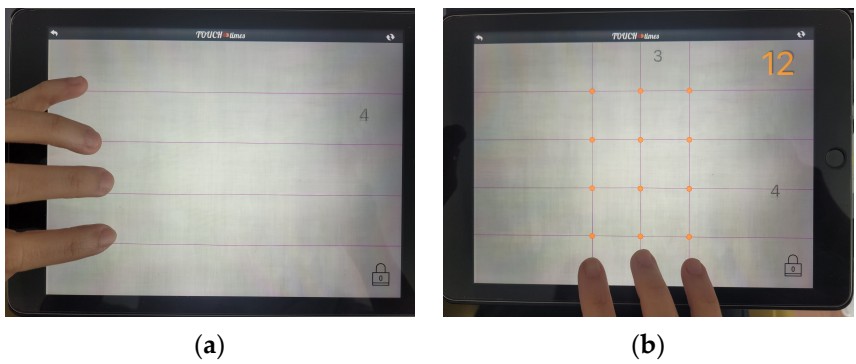

| (a) | (b) |

**Figure 2.** (**a**) Making 4 horizontal lines; (**b**) making 3 vertical lines.

Overall, the two microworlds provide two different approaches and contexts for developing multiplicative thinking in terms of the placement of fingers, the objects that are created, and what they represent and how they can be viewed (see Table 1). In Grasplify, the two hands do different things: one hand creates the unit and the other the multiples of the unit, which are copies of the unit. In contrast, in Zaplify, both hands are doing the same thing. In both worlds, the functionality of their fingers is the same in that they touch, but the outcome of the symbolic relations, as expressed on the screen, is quite different. From

a design perspective, Grasplify is meant to symbolise a unary relation and Zaplify, a binary relation. In Zaplify, the two hands come together in a coordinated simultaneous way to make a grid of points, while in Grasplify, there is a more measured set of steps: choosing a unit, establishing the unit, choosing a number of copies, and establishing the copies into a whole.

**Table 1.** Some notable differences between Grasplify and Zaplify.

|  | **Grasplify** | **Zaplify** |
|---|---|---|
| Placement of fingers | Right and left | Side and bottom |
| Screen objects | Pips and pods | Horizontal and vertical lines making intersection points |
| Multiplication objects | Multiplicand and multiplier; expression (i.e., $3 \times 4$) | Factors; two numbers (i.e., 3 on the top and 4 on the side) |
| Orientation/view | Landscape | Portrait or landscape |

*4.2. Research Context*

To help us explore how students translate between Grasplify and Zaplify, we conducted task-based interviews with four pairs of grades 3–4 students (8–9-year-olds) in the Lower Mainland of British Columbia, Canada. The students were all in the same class, in a French immersion school, with an experienced teacher who had used TT before in her teaching. The teacher had identified the students as representing a diverse range of ability in mathematics. However, none were identified as having learning disabilities. These students all had prior experience using TT in their mathematics classroom while learning about multiplication in both English and French. The interviews were conducted by the three co-authors in only English and lasted between 20 and 30 min. They were conducted in the hallway, during class time, without the presence of their teacher. We video-recorded these interviews to capture their discussion and their actions while interacting with TT. All students were given pseudonyms for this paper: Ian and Fabian; Leo and Rob; Mandy and Tania; Bert and Wayne.

The task consisted of three multiplicative word problems that relate to Ideas of grouping or unitising and array model, which corresponds to the two microworlds. The word problems were read out and shown on strips of paper to students. Students were given the choice to decide which world in TT they would prefer to use to solve the problems. After solving it successfully, they were prompted to switch to the other world to solve the problem or to explain their solution again in relation to the other world. The numeric product value feature of TT was purposefully turned off so that students would not only focus on the answers to the problems but also the multiplicative thinking involved.

1. A bunch of buttons fell on the floor. Nick gathered them in heaps of 5 buttons. He made 8 heaps. How many buttons are there?
2. Aria planted some tomatoes in her mini-garden. She planted them in rows of 5 tomatoes and made 9 rows. How many tomatoes did she plant?

An additional word problem that is based on the fundamental counting principle was included to observe how students associate them to the two microworlds.

3. Colin has 4 hoodies and 5 sweatpants. How many ways can he match the hoodie and the sweatpants?

In addition, during the interviews, students were given multiplicative statements such as $4 \times 3$ or $4 \times 6$ in one microworld and were explicitly asked to create the same statement in the other microworld. The interviewers were also deliberate in probing and sometimes asked direct questions to elicit students' ideas of relating the two microworlds.

## 5. Coordinating between Grasplify and Zaplify

Following Diagne, we take translation to be less about accomplishing a task (i.e., translating Zulu to English) and more about seeing the opportunities to move between

languages—in this case, our languages are the different microworlds of multiplication. We insist on the fact that what is being translated is not a Platonic concept of multiplication, but ways of thinking and acting and feeling. If Grasplify is the world students first encounter, the experience is not a grounding to multiplication; rather, Grasplify provides a locality that involves certain ways of speaking and moving, of seeing and noticing. Grasplify has its own cosmology in which students touch and gesture, in which pips and pods appear on the screen, as well as symbols and expressions. When they move to Zaplify, they are not initially aware of the resonance with Grasplify; rather, Zaplify is a new locality with its own immersiveness. But the resonance emerges through their touch, their perception of the lines, pods, and symbols, and their follow-up interactions.

To convey the fullness of what we intend by "thinking and acting and feeling", we have selected three episodes from the same pair of students that feature significant translations between Grasplify and Zaplify. Drawing on the same pair of students enables us to provide the readers with a sense of how their multiplicative thinking evolved over time. We present these first in the subsection "Three cases of translation with Ian and Fabian". In the next subsection, "Patterns of translation", we focus on the broader patterns of translation that we observed across the four pairs of students during the interviews. We also highlight some of the untranslatables we noticed.

*5.1. Three Cases of Translation with Ian and Fabian*

Case 1: Translating the identification of units.

In answering the button question in Grasplify, Ian touches five pips simultaneously with his left hand and then eight pods sequentially with his right hand. He then counts the number of pips in a pod by pointing with his right hand and holding out all five fingers on his left hand at once and then using his fingers as units of five, skip counting by 5 s (he lifted all five fingers and then three more) to obtain 40. The multiplicative expression $5 \times 8$ is at the top of the screen but the product is not showing.

The interviewer asks the pair to use Zaplify to answer the same question. Ian touches five horizontal lines with his left hand and sequentially touches eight vertical lines with his right hand. He then counts the number of intersection points along the bottom (there are eight). He counts the number of intersections along a vertical, obtaining five. The interviewer says, "So there's five times eight". Ian then says, "You can just count the buttons". Ian counts along the bottom line, one by one to eight; Fabian counts along the vertical to five, and then Ian goes to the top row and counts by 5 s, not on his fingers this time, but by pointing and counting the buttons along the top row. He says "40" and seems satisfied, perhaps because it matched the answer from Grasplify.

In the shift from Grasplify to Zaplify, several aspects relevant to translation occurred. First, Ian used the same hand gestures, touching five times simultaneously with his left hand and then eight times sequentially with his right hand. The screen actions therefore translated directly. Second, in both cases, the pair also sought to find the value of the product, which they did by skip-counting. However, whereas the number of pods was not explicitly named in Grasplify, it was in Zaplify, where the pair counted both the number of horizontal lines and the number of vertical lines. The status of the factors and therefore the way of finding the product did not translate directly. Specifically, the units of five in Grasplify seem more evident than in Zaplify. We note that the problem context of buttons becomes relevant in Zaplify, since Ian evoked having to count the number of "buttons", while pointing to the points of intersection. This observation highlights the fact that the pair was engaged in a double translation, from word problem to Grasplify to Zaplify.

Case 2: Translating between points and lines.

While working on the tomato problem, Ian immediately chooses Grasplify and placed five fingers on the left and then touched two fingers at a time, five times, configuring the 10 pods into five rows of two pods. He then reads the problem again, touches once more with two fingers to make 12 pods in total. Fabian reads the question out loud. They both start to drag pods to the trash and accidentally lift all the pip-fingers, so they have to start

again. Ian then makes five pips and 10 pods, drags one to the trash (Figure 3a). Fabian counts the number of pods. Ian presses the array button momentarily and releases it. There is a silence, then the interviewer asks, "Can you tell me what's going on?" Fabian says, "there are five rows". Ian then touches the array button and holds it for 34 s (Figure 3b). The interviewer asks, "how many rows are there? If you see this as a row", pointing at the top row in the array. Fabian says, "If you were to see this as a row, five. Wait". Ian says "nine". Fabian starts to count the rows, but Ian repeats "nine" and Fabian says "yeah nine". When asked if there was a reason for their choice of the Grasplify world, Ian says, "because you can make it in a row unlike the other one". The interviewer then suggests they try the problem in Zaplify.

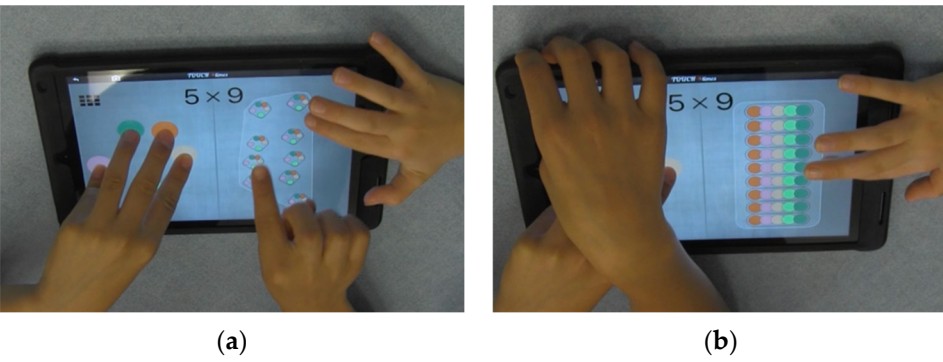

(**a**)                                 (**b**)

**Figure 3.** (**a**) Five pips and 9 pods; (**b**) holding the array button.

In Zaplify, Ian created five horizontal lines with his left hand and then nine vertical lines with his right hand. The interviewer then asked, "Ok, what do you think of that, what would your interpretation be, of nine rows of five?" Ian counted exactly like he had done with the buttons problem, along the bottom from left to right by pointing to each intersection point. He then counted the vertical lines and quickly he said, "Five. Yeah, there's nine rows by five". Fabian said, "Yeah there's five" and he moved his finger up the vertical line and then used a chopping gesture (Figure 4a) above the bottom row, while saying "there are nine". The interviewer then commented, "these look like rows [pointing to the horizontal lines], right? Do you think this microworld is better?" Ian said, "probably" and then elaborated, "because the rows are actually like even rows (gesturing along the horizontal row), unlike the other (fades off). It's like automatically even, it's faster", moving his right hand horizontally over the row back and forth. Fabian said something about Grasplify, of which most was inaudible, but he tapped his finger three times on the table (Figure 4b). Ian continued, "In the other one [Grasplify], you have to count each one in the little [trails off, but referring to the pods]".

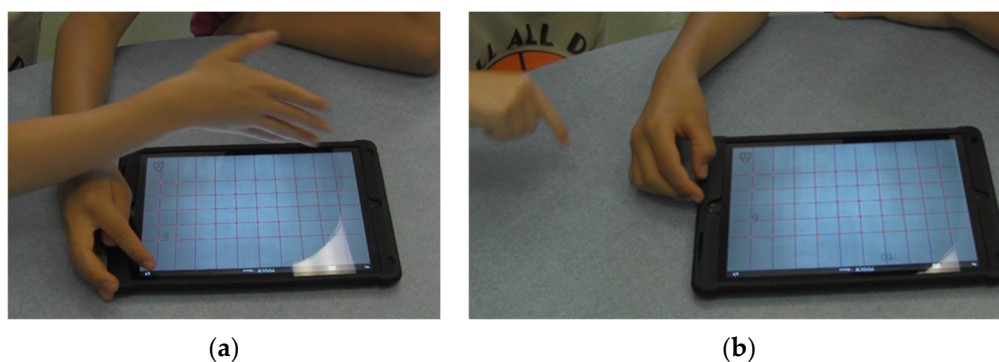

(**a**)                                 (**b**)

**Figure 4.** (**a**) Chopping gesture; (**b**) counting gesture.

In Grasplify, modelling the tomato problem took several tries. The making of five pips (using a simultaneous left-hand tap) was immediate, but making the pods involved some trial and error—they first made 10 pods, then 12, and then finally 9. The mention of rows in

the problem seemed to make Ian want to create rows of pods. When the pair shifted to the array view, this changed their perception of the rows, which they were first seeing as the columns of the array. The translation from the word problem to Grasplify was therefore not transparent. However, when the pair shifted to Zaplify, Ian quickly made five horizontal lines and then nine vertical lines. He noticed that Zaplify produces the horizontal lines, which are the rows of the problem, "automatically", which is indeed "faster". Whereas Grasplify drew Ian's attention to the tomatoes, through the making of pips, Zaplify seemed to draw attention towards the rows, allowing Ian to coordinate the horizontal lines on the screen with the rows of the problem.

When describing Zaplify, the pair started to make gestures that were specific to the rows and columns, including chopping and waving gestures that not only express the rows and columns, but also seem to evoke these individual rows and columns as units. Whereas in Grasplify, the unit is a pod, which as Fabian suggested, through the tapping gesture, that you count, in Zaplify, the unit is a line, which includes the relevant number of intersection points. In other words, the unit, which was a count in Grasplify, translates into a chopping or waving gesture in Zaplify.

Doing the problem in each microworld highlights a different aspect of it, as well as a different object in the microworld. This aligns with Diagne's idea of seeing the opportunities in both worlds, so that translating involves switching between tomatoes and rows. In this case, the translation is the awareness of the various features in the two worlds. In Ian's words, in Grasplify, "you have to count each one inside the pods", while in Zaplify, "the rows are actual rows". The two microworlds have different local contexts and suggests to Ian that in one the focus is on tomatoes as objects, and in the other, rows are the dominant objects. These can be seen as two ways of planting tomatoes; one can dig the troughs for the seeds and then put in the seeds later, in which case the focus will be on the trough, or one can count the seeds and sprinkle them according to how many seeds there are.

Case 3: Translating between Grasplify arrays and Zaplify.

In the previous two episodes, we examined the translations that occurred when Ian and Fabian went from one world to the other as they modelled a word problem. In this episode, we want to focus on an episode in which the interviewer asked the pair to directly compare elements of each world. This provides insight both into how some visual element-to-element translations are more challenging and how provoking these forms of translation require direct and explicit questioning.

After completing the second problem, the interviewer asks Ian and Fabian to explain how the rows and columns of Zaplify related to the pips and pods of Grasplify. They say that the pips were the intersection points and the pods are the rows. They also circle the rows in Zaplify to demonstrate how they correspond to the pods containing pips in Grasplify. When asked where the rows in Zaplify could be seen in Grasplify, Ian and Fabian make four pips and five pods, then press the array button. They then draw horizontal and vertical lines between the pips and create additional dots where the lines intersected (Figure 5a). The interviewer prompt the students again, "can you think of these discs as the orange connectors?" The students then draw vertical lines that go through the pips, but the horizontal lines do not (Figure 5b). The interviewer directs their attention to the number of horizontal lines drawn compared to the number of rows of pips in the array and asks if the two should line up. Eventually, the students draw five horizontal lines that go through the five rows of pips (Figure 5c). When asked to explain how their annotations helped to make it look like Zaplify, Ian says, "in between the lines, there are dots and that's how you can tell how many dots there are".

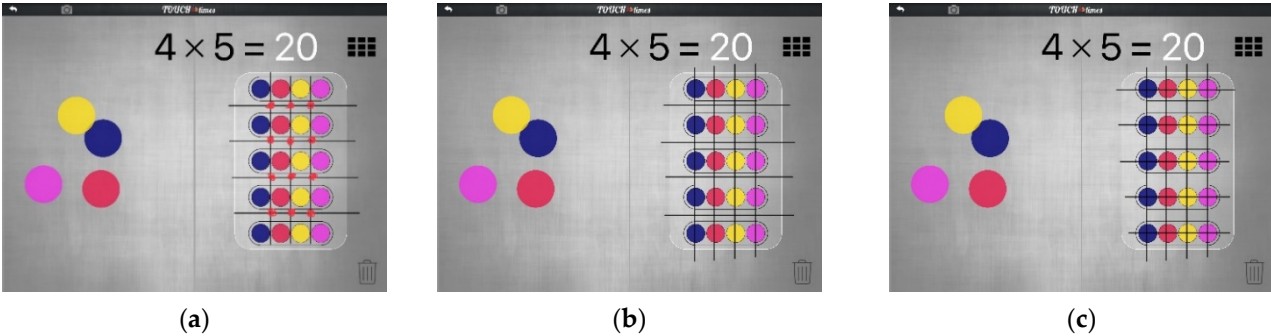

**Figure 5.** (**a**) Initial attempt; (**b**) second attempt; (**c**) final attempt.

This episode highlights how non-trivial the visual translation across the two microworlds was for the students. They were fluent when it came to translating a word problem containing two factors from one world to the other. However, the visual signs (pips, pods, intersection points, and vertical and horizontal lines) seemed to be specific to each world. It was only with much probing by the interviewer that the horizontal and vertical lines and intersections of Zaplify could be seen as annotation lines going through the array of pips in Grasplify.

### 5.2. Patterns of Translations

While the previous section had focused on specific cases in which we noticed interesting or unusual instances of translation with Ian and Fabian, this section provides some overall patterns across the four pairs of students, highlighting common translations, as well as untranslatables. We begin with the word problems and discuss how students went about enacting them, noting which microworld the students chose to use first and what they did and said when they enacted the problem in the other microworld.

#### 5.2.1. Button Problem

For the button problem, two pairs chose to first solve it in Grasplify, and two pairs chose Zaplify first. Although they did not all start with the same microworld, all the pairs started by tapping five fingers on the left (to make pips in Grasplify and horizontal lines in Zaplify), then tapping eight fingers on the right or the bottom (to create pods in Grasplify and vertical lines in Zaplify). These finger placements and the sequential order are aligned to the context of the problem of having five buttons in each heap and making eight heaps. Thus, there is a translation from the problem context to where the students placed their fingers and in which order they placed them. Of the three pairs that also solved the problem in the other microworld, two pairs also started by tapping five fingers on the left, then tapping eight fingers on the right or the bottom. We can therefore say that these two pairs haptically translated the action, both in terms of the sequential order (5 and then 8) and in terms of the position (left, then right/bottom) from one microworld to the other. This is interesting because the problem could also have been solved in Zaplify by placing eight fingers along the left side and five along the right or bottom. In other words, sequential touching translates even when order does not matter.

In one case, the haptic translation pertained also to the way the factors were created. Bert and Wayne made 5 × 8 in Zaplify by first making horizontal lines followed by five vertical lines, followed by another three vertical lines (Figure 6a). They tried to find the product, first saying 13, then 80, and then finally, 40. When asked if they can do the problem in Grasplify, Bert put five fingers on the left, Wayne put five and then three fingers on the right (Figure 6b), exactly like he had done in Zaplify. The breaking down of eight into five and three may feel haptically more natural than tapping eight fingers all at once.

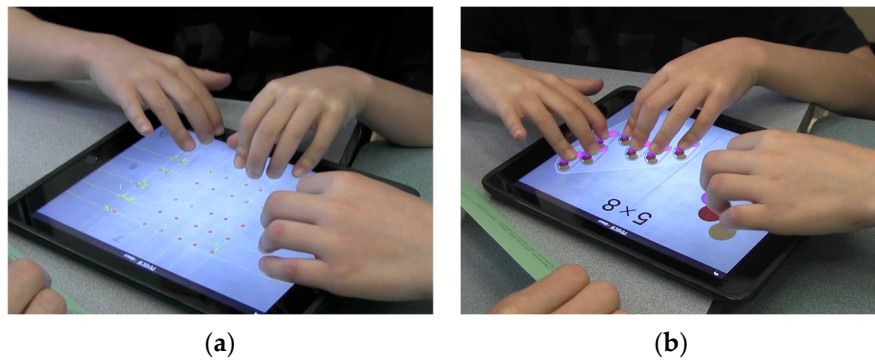

**Figure 6.** (**a**) Finger placement in Zaplify; (**b**) finger placement in Grasplify.

The other pair of students who solved the problem in a second world, going from Zaplify to Grasplify, did not translate their actions. Before using Zaplify, Leo had stated the product (40) out loud. While he was doing that, Rob had already opened Zaplify and had placed five fingers on the left (refer to Figure 7a, where the fingers are positioned horizontally rather than vertically). Leo pressed the lock button and made eight sequential taps on the bottom. When asked whether they could do the problem in Grasplify, Leo first placed five pip-fingers on the left side of the screen simultaneously (Figure 7b), then using his right hand placed three more fingers on the left side, while Rob made five pods sequentially, thereby obtaining 8 × 5 (Figure 7c). When asked how Grasplify shows the buttons and heaps, Leo explained that the heaps are on the right side and the buttons on the left, while re-making the product as five pips and eight pods.

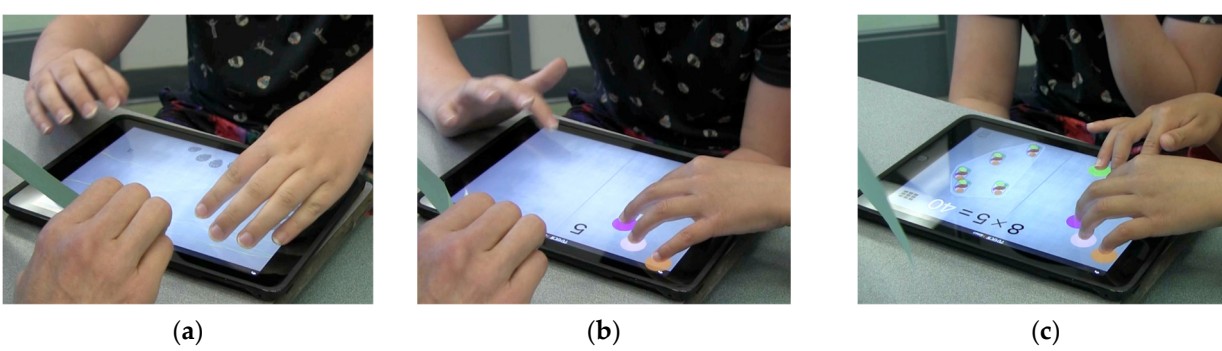

**Figure 7.** (**a**) Positing the hand horizontally instead of along the left edge; (**b**) making 5 pips in Grasplify; (**c**) making 8 pips and 5 pods in Grasplify.

The order and the placement of the fingers were initially reversed, when going from Zaplify to Grasplify, which suggests that the translation is not occurring haptically and is instead focused on the value of the factors. We note also that the Grasplify solution does not reflect the problem statement, which involves eight heaps of five buttons. When Leo re-made the product, however, he reversed the order again. It is therefore the problem statement itself that changed the translation. After Leo finished making the product, Rob remarked that Grasplify is better because it shows the groups more clearly than Zaplify. When asked if it is possible to see the groups in Zaplify, Leo said "no". Rob changed to Zaplify, made 3 × 3, and then said that you cannot. But then Leo changed his mind, saying "in the vertical and the horizontal lines", gesturing along each of the lines. They both agreed, however, that it is not as obvious. We see here Leo making a translation from the pods to the lines, but recognising the translation is not perfect.

5.2.2. Tomato Problem

For the tomato problem, again, the same two pairs first solved it in Grasplify and the other two pairs first solved it in Zaplify. Although they did not all start with the same

world, three pairs started by making five taps (pips in Grasplify and horizontal lines in Zaplify), then nine taps (pods in Grasplify and vertical lines in Zaplify). In Grasplify, these finger placements and the sequential order are aligned to the context of the problem of having five tomatoes in each row and making nine rows, but in Zaplify, they do not reflect the context of the problem as there are five rows of nine tomatoes instead. Of the three pairs that also solved the problem in the other world, two pairs made the exact same tapping actions. Again, there was a haptic translation between the worlds both in terms of the sequential order (5 and then 9) and in terms of the position (left, then right/bottom).

Mandy and Tania proceeded differently. They started in Zaplify. Mandy made one vertical line first, then pressed the lock button. She created four more vertical lines sequentially, after which she created nine horizontal lines sequentially. But when the interviewer asked her to explain what she had done, she switched to Grasplify and said, "this is easier". In Grasplify, Mandy made five pips simultaneously, then created pods sequentially, configuring them into four rows of two pods and then an additional pod underneath the four rows. She explained that the solution is 45 because the previous problem was $5 \times 8$ (for the button problem) and it was 40, so she would add 5 to 40. In this case, there was no haptic translation and interestingly a preference for using Grasplify, perhaps because it enabled the direct creation of objects (the pips, which could be interpreted as the tomatoes) whereas Zaplify only produces objects (the intersections) once there are both horizontal and vertical lines. Therefore, translated here are the tomatoes of the problem into the pips of Grasplify.

Indeed, when asked to explain what the pips and the pods represent, Mandy replied, "these dots (pointing at the pips) are the tomatoes, and these are the nine rows (pointing at the pods)". The interviewer asked the pair to go back to Zaplify. Again, Mandy first created one vertical line and pressed the lock button. She then made four more vertical lines sequentially. The interviewer interrupted and asked, "What do the five vertical lines represent?" Mandy replied "tomatoes". She then created nine horizontal lines. Although these actions are not the same as her actions in Grasplify, they are consistent with her initial attempt in Zaplify and also reflect the problem context of nine rows of tomatoes with five in each row. Thus, it seems that there is a translation from the problem context of nine rows (of tomatoes) to the nine horizontal lines and five tomatoes in each row to the five vertical lines.

When asked to explain where the rows and tomatoes are, initially she said, "like five here (using her index finger to swipe up and down vertically)", then she hesitated and said, "no, five here (using her index finger to swipe left and right horizontally) and nine here (using her index and third finger to swipe up and down vertically). And the nine represents kind of like the rows (used her right hand to swipe left and right horizontally) and there will be five on each row". Hence, in Grasplify, the units are seen as points, both pips and pods, while in Zaplify, the units seem to have been translated to lines, as implied from the switch Mandy made from a pointing to a swiping gesture.

For the last pair, when the interviewer read out the problem, Wayne said, "I just look at the numbers and times". As soon as the interviewer said, "how many tomatoes", Wayne answered 45. When asked which microworld they would use, they quickly selected Zaplify, with Wayne saying "there's rows in this one". Bert tapped three fingers on the side and the interviewer said, "I don't see the tomatoes". Bert placed three fingers on the bottom and said, "there are the tomatoes", pointing to the intersection points. Bert said, "you get the tomatoes after a couple stages of plants". Meanwhile, they added more lines on the screen, until there were eight columns and five rows. There is no comment on the fact that the problem talks about nine rows. The interviewer did not ask the boys to solve the problem in Grasplify, so there is little to say about translation between the microworlds. However, it is clear that there is a translation from the problem context of rows (of tomatoes) to the choice of Zaplify and that the tomatoes were translated into intersection points, with an awareness that these emerge after the creation of horizontal and vertical lines—Bert even introduced a new metaphor of "stage of plants" to describe the temporal emergence

of the tomatoes/intersections. This can be seen as a creative translation that draws on the language of Zaplify to elaborate the context of the problem.

### 5.2.3. Combinatorics Problem

Finally, we discuss the combinatorics problem, which only two of the pairs attempted. Both started solving the problem without using TT and, when prompted, chose Grasplify. After hearing the problem, Leo immediately moved his hand as if he was about to use Grasplify, but pulled it back to touch the table, saying "I have one hoodie (extending one left-hand finger) and then I have like five sweatpants (putting his five right-hand fingers on the desk). And that one hoodie can go with five of them (moving his left-hand finger back and forth to each of the right fingers) so [...] and then there are three other hoodies so you just do it with those (gesturing back and forth again)".

When prompted to think about the same problem in either Grasplify or Zaplify, Leo chose Grasplify, which was already open. He created one pip, referring to it as one hoodie and then five pods, which he called the sweatpants (Figure 8a). Similar to the gesture he made before, he moved his pip-finger back and forth, dragging the pip to each of the five pods and said, "you can pair one (pip) with each of them (pods)". He created three more pips for a total of four and said, "and like you can do it four times to get twenty styles" (Figure 8b).

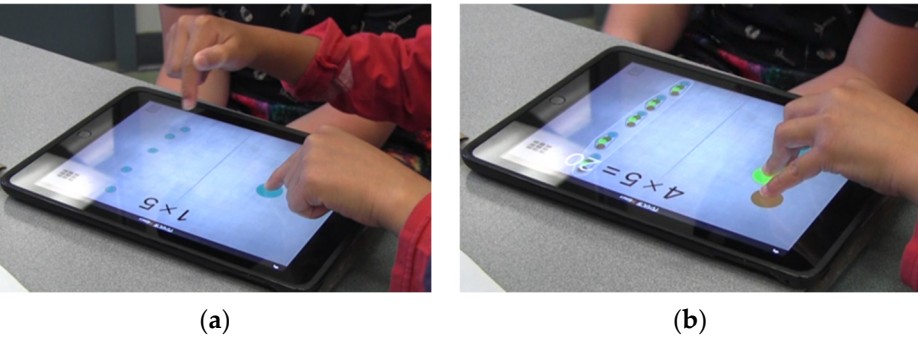

**Figure 8.** (**a**) One hoodie (pip) and 5 sweatshirts (pods); (**b**) 4 hoodies and 5 sweatshirts.

The interviewer then asked them to do the problem in Zaplify. Leo said, "Okay, so you like have one of them (creating a single horizontal row, then touching the lock button) and then you have five (touching five times sequentially, making vertical lines, as in Figure 9a)". The interviewer said, "So this is one combo?", pointing to an intersection. Leo answered, "Yeah, and you can see all the combos (gesturing along the row, back and forth, as in Figure 9b, to all the red dots and then touching three more times on the horizontal). And you can see four, see, you can see all of the combos and it's like (showing the lines too, as in Figure 9c) so you know, which ones... [trails off]... so you know which pants go with which hoodies". Leo emphasised the word "know", highlighting his awareness that each intersection represents a specific combo. This explanation with his fingers, gestures, and words does not draw on the visual output of either Grasplify or Zaplify, but does draw on the haptics of using TT.

Leo solved the problem with his hands, then with Grasplify. In both cases, he used a back-and-forth, one-to-one matching gesture to connect the articles of clothing with each other. This dynamic gesture is analogous to taking a hoodie and moving it from one pair of sweatpants to the next to see which is the best combination. But there is a translation that occurs when moving to Zaplify. There is no one-to-one dynamic matching in Zaplify. Zaplify in this problem is less about getting dressed and more about laying out all the possibilities. This is impossible to do in real life, since you only have one green hoodie, so you cannot lay it down and pair it with all the different sweatpants at the same time. In this way, Zaplify does not have a real-life analogy in this problem. Rather, Zaplify presents a static expression of all the possibilities, which Leo acknowledged when he said, "so you know which pants go with which hoodies". He drew upon the notion of lines, as

seen in Figure 9c, expressing that the lines are these links, which is no longer his one-to-one matching. Each horizontal line is a hoodie and each vertical line is a pair of sweatpants. This translation between one-to-one matching to an acknowledgement of all possibilities at once can be seen as a move from a unary relationship, where factors play different roles, to a Cartesian product relationship. In a Cartesian product, each number in one set is listed out with each number in another set, or, in this case, each line in the vertical is matched with each line in the horizontal.

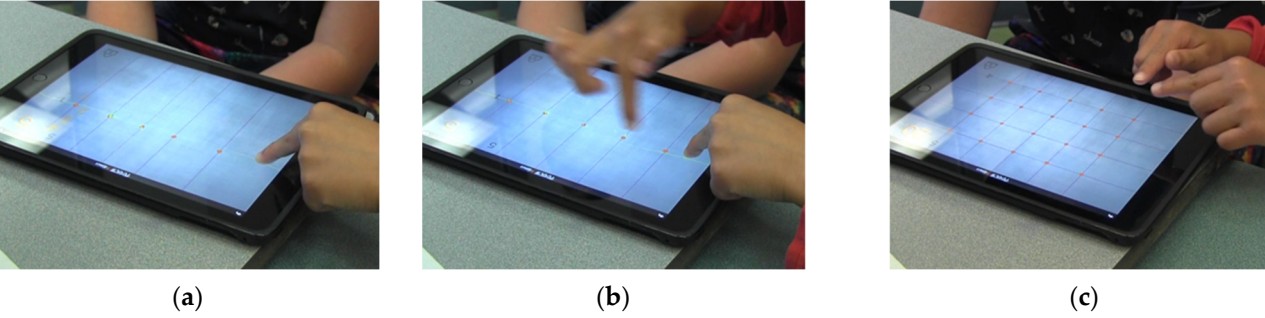

(**a**)　　　　　　　　　　　(**b**)　　　　　　　　　　　(**c**)

**Figure 9.** (**a**) Making 1 horizontal line; (**b**) making 5 vertical lines; (**c**) showing intersections.

When Bert and Wayne were presented with the problem, they were unsure how to proceed. The interviewer simplified the problem to two sweatpants and used strips of paper to represent the hoodies and sweatpants and also did a one-to-one matching gesture to pair each of the hoodies with one sweatpants. The pair proceeded to solve the problem (with two sweatpants) in Grasplify. They created two pips and five pods. When asked to explain what the pips and pods represent, Bert said, "We can literally match any of them like this", while dragging one pod from the right to meet the two pips on the left (Figure 10), which is similar to what Leo did. There is a translation from the interviewer's one-to-one dynamic matching using strips of paper to the one-to-one matching of the pods (representing hoodies) to the pips (representing sweatpants) in Grasplify.

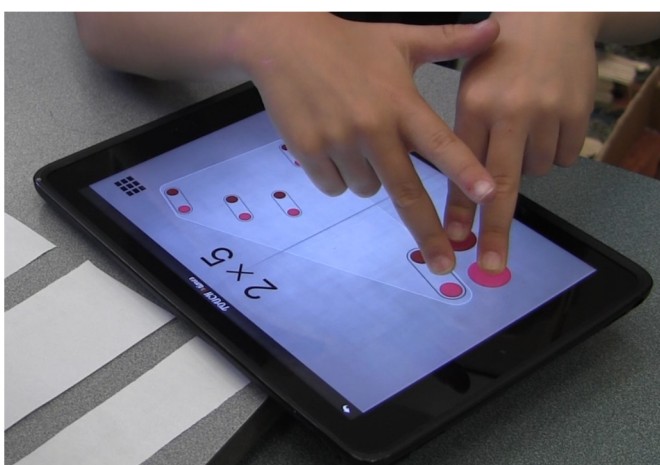

**Figure 10.** Matching pod to pips.

### 5.3. Follow-Up Questions

After working through the three questions, the interviewers explicitly asked the students to compare the pips and pods of Grasplify with the lines of Zaplify. For Bert and Wayne, the interviewer had created $4 \times 3$ in Grasplify and asked them to make the same product in Zaplify. They haptically translated, with Wayne making four horizontal lines and Bert making three vertical lines. When asked to explain how the lines in Zaplify are the same as the pips and pods in Grasplify, Bert replied, "well, it's dots and groups". Wayne added, "it just doesn't have lines... wait, are those groups of four?" The interviewer replied,

"yes, there are groups of four. So, are there groups here (referring to Zaplify)? Is there a way to see groups here?" Bert said, "well, ya, like if you draw circles around all of these... This is a group of three (circling gesture around the topmost row of 3 points)". The interviewer asked, "are there any groups of four?", to which Bert replied, "like this, (circling gesture around the leftmost column of 4 points, as outlined in a black frame in Figure 11a)". Wayne added, "like squares are groups of four (gesturing in a shape of square around 4 points forming a square, as in Figure 11b)".

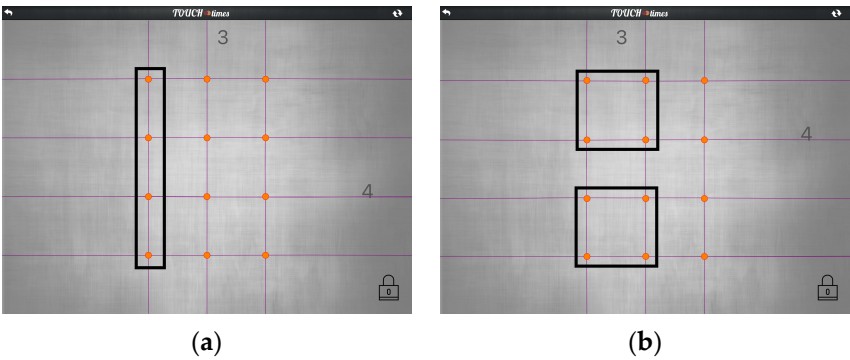

(a)         (b)

**Figure 11.** (**a**) Four points in a column; (**b**) 4 points form a square.

Ian and Fabian were given the same situation, in which the interviewer made 4 × 3 in Grasplify and they were asked to describe where the pods are in Zaplify. Ian did a circling gesture around the topmost row of three points and responded, "here, the rows?" The interviewer asked, "but when I look here, I see three. And if I am looking for pods, how many are in a pod?" Both students replied "four", and Ian quickly cleared the screen and created three horizontal lines and then four vertical lines. Then, he did the same circling gesture around the topmost row of four points and said "here" to illustrate where the four pods could be seen.

For Mandy and Tania, the interviewer created 4 × 6 in Grasplify and asked them to make the product in Zaplify. Tania created one vertical line, pressed the lock button, and then made three more vertical lines. She then made six horizontal lines. When asked to explain where the four pips are in Zaplify, Tania replied, "oh, here, (swiping gesture along the topmost row) . . . I mean, these things (swiping gesture up and down along the columns with index finger, as in Figure 12a, first followed by similar swiping gesture with her hand, as in Figure 12b, going from right to left)", referring to the four vertical lines. And when explaining about the six pods, Mandy swiped from left to right along the bottom row and repeated the same gesture for the other five rows.

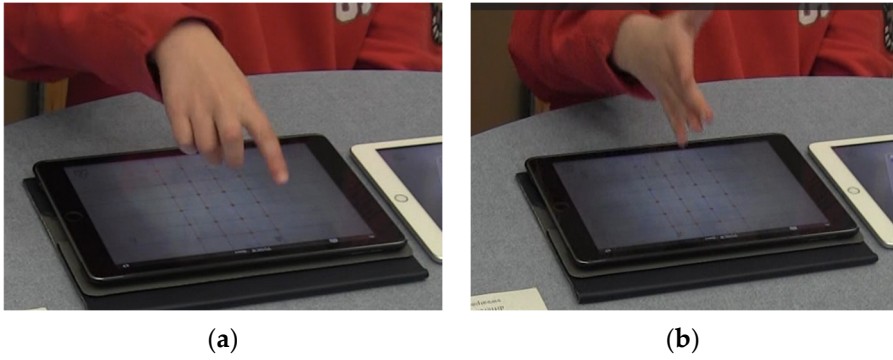

(a)         (b)

**Figure 12.** (**a**) Swiping gesture with finger; (**b**) swiping gesture with hand.

The way the students expressed pips and pods in Zaplify varied, although most pairs were more inclined to see the groups as rows rather than columns. Indeed, Ian and Fabian reset their screen in Zaplify, which had three rows and four columns, and made four rows and three columns, so they could circle four rows instead of four columns.

Thinking with Diagne's idea of translation, which emphasises the values that are at play in different cultures and the effect these have on what is translated and how, we were especially attentive to preferences that the students articulated with respect to each microworld. These preferences were sometimes related to the problem itself, but other times to the appearance and the functionality of each world. For example, when introduced to the tomato problem, Rob and Leo already had Zaplify open. After listening to the problem, Rob changed to Grasplify. However, Leo said that "since the problem talks about rows and there are already rows in Zaplify, it would be easier to understand". But they made five rows and nine columns, even though the problem states that there are nine rows. When asked whether they could do it in Grasplify, Rob said it would be easier. Leo made five pips simultaneously and Rob made nine pods sequentially. He said, "I don't know, I just think it's more helpful to see these [gesturing to the pods]". Rob mentioned several times over the course of the interview that he preferred Grasplify for the way that it showed the pods, but It was particularly Interesting In this problem, which was explicitly about rows. We interpret his preference for Grasplify less in terms of whether it models the problem better than in terms of providing him with a better way of understanding the meaning of multiplication. In contrast, Leo mentioned his preference for Zaplify on two occasions. The second time, described in this paragraph, was related to understanding the problem. On the first time, which was for the button problem, Leo said that Zaplify was quicker and easier, because you can use the lock button and not have to hold a hand on the screen.

### 5.4. Untranslatables

During our interviews, we sometimes probed the students on their actions and choices. For example, Bert and Wayne had made $4 \times 3$ in Grasplify (Figure 13a) and the interviewer asked, "So is that the only way you could do it?" Bert said, "Not really, you could switch them around". The interviewer asked, "Can you show me what you mean?" They immediately made three pips and four pods (Figure 13b). The interviewer then asked, "So why can you do it in two different ways here [pointing to Grasplify], but you don't do it in two different ways here [pointing to Zaplify which is showing $4 \times 3$ on a different iPad]?" Wayne said, "Is it because you tried to make them harder?" This points to an untranslatable, that you can switch in Grasplify, but not in Zaplify.

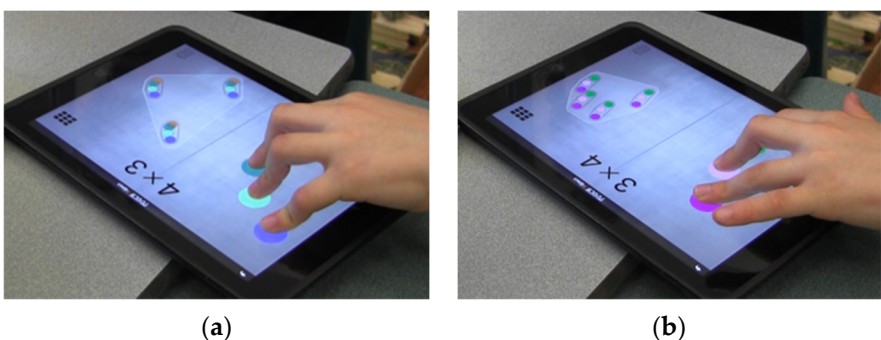

(**a**)        (**b**)

**Figure 13.** (**a**) Making $4 \times 3$; (**b**) making $3 \times 4$.

In Grasplify, the students switched hands quite quickly (we showed an example earlier of Leo switching from $8 \times 5$ to $5 \times 8$). However, we never saw them switch factors in Zaplify. We hypothesise that it is because the visual display changes so much in Grasplify, so that $4 \times 3$ is seen as being different from $3 \times 4$ (though also being the same). Also, the two sides of Grasplify host different kinds of numbers: one side is the unit, the other side is the number of copies of the unit. For both of these reasons, switching sides seems to make a difference for students. In Zaplify, however, the difference is not recognised. Not only would the grids look very similar when the horizontal is switched with the vertical, but also the two sides (left and bottom) create the same kind of objects (the two factors).

Of course, it is possible to switch hands in Zaplify, but no student ever did. We interpret this as an awareness they have of the symmetry of Zaplify, that is, that placing 7 fingers along the bottom and 5 along the side will produce "the same" as placing 5 along the bottom and 7 along the side. Moreover, we suggest that by *not* switching, they are showing a sense of hospitality in Zaplify, by respecting its particular way of showing multiplication and not forcing into it the ways of working in Grasplify.

Related to this issue of switching, the only time we heard the students comment on the difference in Zaplify was when Leo was asked which world was better for showing that 4 × 5 is the same as 5 × 4. He first chose Grasplify, but then switched to Zaplify. He created 4 × 5 and described "there are four dots, five times". He hesitated for five seconds and then turned the iPad 90 degrees, saying "When you look at the portrait, you can see it, (he then turns the iPad 90 degrees), when you look at landscape you can also see it". We note that in this explanation there is no switching of hands, only a change of perspective.

We saw a similar untranslatable when we asked Ian and Fabian to vary the button problem, so that there would only be four heaps (instead of five). In Grasplify, Ian released one pip-finger, while Fabian started to drag one pod into the trash bin. When Ian asked him why he was doing that, Fabian replied, "because four buttons" and Ian responded, "four buttons in each pile", to which Fabian agreed. Subsequently, when the interviewer asked about decreasing the number of buttons in each heap again, they were quick to lift a pip-finger. In comparison, when asked to increase the number of buttons in each heap in Zaplify (from five to six), they spent more time discussing where they should tap, the left side or the bottom. They had made five horizontal rows and eight vertical ones. Ian tapped once at the bottom (creating nine vertical lines) but soon realised that it was incorrect. Fabian tried to help by tapping on the top of the screen to add one more horizontal line and said, "that's six", but Ian said, "but that's nine". Eventually, Ian cleared the screen and restarted. He made six horizontal lines, then locked the screen, and made eight vertical lines. When asked, "what if there were seven buttons?", Ian made one more vertical line but realised that it was incorrect and restarted. We see this as a similar untranslatable because it seems related to the fact that the two quantities play more distinct roles in Grasplify, which seems to make it easier to vary the problem, while the two quantities in Zaplify are harder to distinguish. In other words, the distinctness of Grasplify cannot be expressed in Zaplify.

## 6. Discussion and Conclusions

We have provided an in-depth and systematic account of the work of four pairs of students as they engaged in tasks that explicitly invited them to move from one microworld of multiplication to another in *TouchTimes*. In our analysis, we were interested in how the students translated across microworlds, wanting to pay attention to the various forms of experience at play, including the actions, the interpretation of visual signs, the handling of the quantities and the product, and the feelings associated with using each world (efficiency, ease of understanding, comfort, etc.).

We identified very strong haptic translations that seem to pertain whether the students were moving from Grasplify to Zaplify or vice versa. We see this haptic translation as meaningful in relation to multiplication because it centres the fact that there are two quantities involved (and not just one, which is often the case when students see multiplication as repeated addition). We also note that while Zaplify could handle simultaneous input (i.e., three vertical and four horizontal lines all-at-once), the students most frequently did one quantity first. This may arise from both the sequential order of Grasplify (in which the unit must be created first) as well as from the sequential appearance of the factors in both written and oral forms of the problem statement. An exception was a Zaplify-specific action, in which one line is created in order to use the lock button and then the other lines are made afterwards, which leads to a unique haptic expression that was not translated into Grasplify.

Overall, we see translation between worlds, between problems, between pairs of students as helpful in seeing multiplication as an open concept that is continually being

adjusted [5]. These adjustments are not solely cognitive, but in Diagne's sense, are slow co-ordinations of difference coming together, through touching and counting, intentional actions such as tapping on the table to justify a point, as well as new kinds of gesturing. In observing students move between microworlds, we were able to follow how new expressions and patterns emerged in the shift from one to the other. What we found helpful was seeing how previous expressions were elaborated in new contexts and noticing how the new microworld supported these new expressions. For example, counting was something Ian and Fabian relied upon in both microworlds, counting pips in Grasplify or intersection points in Zaplify. However, in one instance, after Ian counted the sides of a 9 by 5 grid in Zaplify, Fabian used a cutting gesture while saying "and then there's nine". The significance of Fabian's gesture is that it introduces the notion that unitising can be enacted differently depending on the context and also by expressing the unit in various ways, centralising its significance.

Evidently, the choice of tasks influenced how translation occurred, but other material contexts also mattered. For example, in the combinatorics problem, the interviewer utilised the strips of paper that were present to help Bert and Wayne in their translation of the word problem to the TT microworlds. It was an unplanned and unscripted response that was motivated by the situation. The eventual translation is likely to have been different had the interviewer decided to do otherwise such as drawing on the paper to illustrate the one-to-one matching (that may appear more static).

Translation, however, is not always focused on development towards a new understanding. We see the notion of translation as holding both microworlds simultaneously. For example, when Zaplify is open and $9 \times 5$ is showing, Fabian introduced his first cutting gesture, and then 15 seconds later, when his partner, Ian, was suggesting Zaplify is better at representing rows for the tomato problem, Fabian suggested that Grasplify is just as good and tapped on the table with his finger three times. This helps us see that translation is about holding different conceptions depending on the context and not leading towards one unified conception. What works in one microworld is carried over to the other world, not to be lost or to be further developed, but to exist as a backdrop while motivating new interpretations and configurations. What we have learned through this research is that there are fluid translations such as where and how students touch the iPad screen, but when an action is not a simple translation, students make adjustments and co-ordinations, leading to a more robust sense of multiplication. In the case when Fabian tapped three times on the table, it also seems clear that preference matters. Fabian preferred Grasplify even though Ian had just indicated that he thought that Zaplify was better for the tomato problem. Ian had originally suggested Grasplify was better but when asked after using Zaplify, he said Zaplify was "probably" better. Diagne notes how translations can occur in one's beliefs. In this case, we see the difference between Fabian and Ian in terms of what each of them believes Grasplify and Zaplify and their functionalities can do in different contexts.

Related to the question of preference, we realised the importance of asking questions that require students to make distinctions between the two microworlds, instead of only focusing on drawing similarities and making connections, to avoid negating the plurality of ways of understanding multiplication. For instance, the question asking which microworld is better for understanding the commutative property of multiplication encouraged the students to put the two microworlds in conversation, but also respecting the differences each affords.

Through our analyses, we identified one significant untranslatable, which relates to the asymmetry of Grasplify in contrast to the symmetry of Zaplify. Specifically, the factors of Grasplify are quite distinct, especially visually, while they are not so distinct in Zaplify, and this seems to affect the way students work with variations of multiplicative expressions. Indeed, this difference between the two microworlds is significant in terms of multiplication and seems to us important to recognise, especially if we want students to not only be aware of different models or representations, but choose them according to the needs or context of the problems they are solving. The only question that proved

this was asked of Leo and Rob, and it was about which microworld makes it easier to see that $4 \times 5$ should be the same as $5 \times 4$. We wonder whether there are other tasks that might prompt students to recognise when symmetry might be important. We also wonder whether designing tasks to evoke untranslatables might be pedagogically valuable.

In addition to untranslatables, we also identified two instances of what we will call creative translations. The first was in the tomato problem, when Bert and Wayne, came up with the idea of the tomatoes growing after the planting, which was a new way of expressing the problem, which was influenced by the use of Zaplify. The second was when Leo and Rob worked on the combinatorics problem and Leo translated his gestures on the table into new ways of seeing and using both Grasplify and Zaplify. We find these two examples important for the way that they speak to the indeterminacy of translation, that there is not a pre-defined set of meanings that are established by the two microworlds; they are open to hybridisation.

In conclusion, we found Diagne's notion of translation apt for the TT research context. For example, the word "line" did not emerge when using Grasplify, but it did so in Zaplify. This is important as it is an example of a new word that has meaning in one world but not the other, yet we would like to see this word as belonging to the plurality of all the meanings of multiplication. Following Diagne, we are less interested in a single dominant or universal language (of multiplication) than in "a world of plurality, weaving together languages and cultures" [23] (p. 21). Therefore, instead of pursuing the kind of vertical translation that vectors every representation or model or microworld to a single meaning of multiplication, we see our work as privileging "lateral translations" (p. 20) that tests meanings across different contexts, different languages, and different people. And as we have shown in our analysis, there is a need for a lateral translation to make sense of the word "line" in Grasplify. We recognise that Diagne is operating within a much broader post-colonial project that critiques the undervaluing of non-European languages (and therefore, non-European thought), while at the same time insisting that we need to think in multiple languages, rather than take refuge in relativism. However, his tropism towards pluralism and valuing of translation has oriented our research by encouraging us to pay attention to both the production of multiple "languages" of multiplication and the meanings that arise through thinking of one language in terms of another.

**Author Contributions:** Conceptualization, S.T., S.C. and N.S.; methodology, S.T., S.C. and N.S.; formal analysis, S.T., S.C. and N.S.; investigation, S.T., S.C. and N.S.; resources, S.C. and N.S.; data curation, S.T., S.C. and N.S.; writing—original draft preparation, S.T., S.C. and N.S.; writing—review and editing, S.T., S.C. and N.S.; funding acquisition, N.S. and S.C. All authors have read and agreed to the published version of the manuscript.

**Funding:** This study was funded by an Insight Grant from the Social Sciences and Humanities Research Council of Canada.

**Institutional Review Board Statement:** The study was conducted in accordance with the Declaration of Helsinki, and approved by Simon Fraser University Office of Research Ethics (protocol code: 20180306 and approval Date: 3 November 2021).

**Informed Consent Statement:** Informed consent was obtained from all subjects involved in the study.

**Data Availability Statement:** The datasets presented in this article are not available because of ethical and privacy reasons.

**Conflicts of Interest:** The authors declare no conflict of interest.

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
