# Peer review of "Learning Multiplication by Translating across Microworlds"

_education, doi:10.3390/educsci14040423_

Round 1

Reviewer 1 Report

Comments and Suggestions for Authors

Thanks for the opportunity to review the manuscript "Learning as translating: Exploring the transitions across two 2 digital models of multiplication". Overall, I think the ideas in the paper make an important contribution to both our understanding of how students conceptualise multiplication, with clear implications for teaching and learning. Although I think the paper could be published 'as is', my only wondering is whether some of these implications for teaching and learning could be made more explicit - although the authors may push back that more extensively elaborating on these is beyond the scope of this particular paper. However, I couldn't help but wonder how student opportunities to map their thinking to the two models, and make connections between the two models, could support aspects of multiplication where students tend to display superficial understanding.

Take, for example, student understanding of the associative property, and, in particular, multiplying by a multiple of 10. As illustrated in this article by Downton et al (https://doi.org/10.1007/s13394-020-00351-w) students, including higher achieving students, tend to approach such problems using 'rules without reason' and manipulate the zeroes, rather than think conceptually/ multiplicatively. I wondered how the use of TT might support students to make sense of the sorts of problems presented in the Downton et al. article? Perhaps further commenting on the implications of the research (from both a practice perspective, and a future research directions perspective) would be worthwhile - however I leave this to the discretion of the authors.

Thanks once again for putting together such a thought-provoking and potentially powerfully practical paper. I look forward to downloading the TT app, and thinking further about the connections between the two models presented. I wish the authors all the best in the future with their important research.   

Author Response

Thank you. We have added a reference to the Downtown article. We have also added a little more text in the discussion relative to pedagogical consequences of our study. We do intend to follow up with another study to engage more fully in the reviewer's suggestion for elaboration.

Reviewer 2 Report

Comments and Suggestions for Authors

The article describes a study of the Touch Times app, which enables users to generate visualisations of multiplicative relationships by placing fingertips on a touchscreen, with four pairs of grades 3-4 children in a French immersion school in Canada. Two modes of the app are trialed, Grasplify – in which fingers placed on the left of the screen determine the unit quantity to be multiplied, and fingers on the right determine the number of ‘pods’ of these units which are displayed on the right - and Zaplify – in which placement of fingertips along the left edge and bottom of the screen generate a grid of crossing horizontal and vertical lines, where the intersectional nodes can be seen as the product. The children were given mathematical word problems designed to provoke multiplicative solutions and asked to solve them first in one mode, then the other. Their responses and dialogue as well as subsequent interviews with the researchers were videorecorded. The authors draw on Diagne’s (2022) theories of ‘learning as translation’, among other theories, to analyse the transition between experiences of responding to the same multiplication question in different modes.

Conceptually the article presents a novel application of Diagne’s theory of learning as translation to a digital learning app, and is particularly interesting in that it compares the experiences of two modes of the same app, applied to the same mathematical questions, highlighting the potential pedagogical value of moving between digital interactive contexts with specific identifiable differences – that is, the hardware, and overall user experience design within the app are shared, facilitating focused comparison of the specific aspects that vary. As the authors highlight, this distinguishes it for example from previous studies comparing digital and real-world contexts.

This article I believe contributes philosophically to maths education research by applying Diagne’s theory of translation, as well as offering sound qualitative evidence of primary-age children’s use of an innovative multi-modal app to solve problems, which supports choosing and switching between two distinct pedagogical models of multiplication, contributing to ongoing comparative research into models of multiplication.

The study itself was generally presented clearly, and well-illustrated, highlighting specific strands of experience – such as gestures, and dialogue – which were compared across the two modes.

Two key areas I would suggest honing overall are:

(1)   Various terms were used in the theoretical framing for what could be translated eg models, representations, artefacts, languages, worlds (and later localities/cosmologies) etc, drawn from the theories referenced, which are from various domains. Though the theoretical discussion was valuable as context, as a reader I would have welcomed a subsequent tighter selection of defined terms synthesised from the referenced theories for the purposes of the paper’s analysis itself. The big (and exhilarating) philosophical leap seems to be from Diagne’s translating languages as visions of the world, to switching between specific modes of the app, and I felt I could have used more conceptual guidance in making this leap. One suggestion would be possibly including briefly the word problem as its own ‘world’, which is then translated into the app experience, for instance being explicit about how the word problem was presented and translated, eg was it read out at all in the pairs, pointed to on paper etc. This may help link linguistic translation to translation between app modes.

(2)   The application of Diagne’s theory I found novel and quite profound in pedagogical terms, in particular the ethical aspect. Diagne is not quoted and my French is not strong enough to be confident in understanding the article referenced, but looking at a recent interview I suspect describing similar ideas of reciprocity she is quoted as saying “if you ask what the translator does when she gives hospitality in her language to what has been thought and created in another language, then you see translation as reciprocity, hospitality, putting in touch.” ( https://www.tandfonline.com/doi/full/10.1080/13556509.2023.2275806 ) These ideas feel relevant to both teaching, and the designing of user experience in apps – the aspiration to make the learner/user feel at home in the environment in which the teacher/UX designer/researcher has dominant power. I would have welcomed slightly more critiquing or reflection in the discussion of how it was to apply Diagne to the context of TT, including anything that didn’t quite ‘fit’, e.g. the moment when researchers had to turn to paper strips felt ‘off piste’ and valuable.

Specific comments:

l.55 “To do so, we bring in recent approaches within the broad umbrella of 54 semiotic mediation, which focus on identifying the ways in which students make semiotic 55 links between different artefacts (including models, but also tools).” – * would welcome a rigorous working definition of artefact/model/tool for this paper, especially when bridging (translating?) quite philosophically and terminologically diverse theoretical source domains. E.g. is language regarded as an artefact/tool in the same sense as an app feature, or gestures, and if so are all these (at least partially) translatable?

l.90-112 * could an illustration help anchor the various multiplicative models described visually? I’m thinking for those reader possibly unfamiliar with maths education customs such as numberline jumps, arrays etc.                                                                                   

l.123 “They studied the different signs that arose while students used 124 both environments, highlighting the different signs that were available in each environ-125 ment and showing how the combination of the artefacts can enable students to engage in 126 a more fulsome interaction with the target concept.” * Given the quite nuanced use of terms like sign and artefacts and concepts, could maybe include example of one of the signs in the reference? Even if in brackets as done in the following para.

l.150 * would welcome example here of artifact/object/context as feels potentially analogous to TT

l.157 “it is not surprising that the majority of the semiotic chains identified in the stud-157 ies by Maffia and Maracci are indeed specific words” *are the specific words semiotic chains, or signs which are then chained together?

l.174 “learning across artefacts” * Is this being used interchangeably with models and worlds?

l184 artefact-situation” *would welcome defining this, eg is this a “world”?

l.192 “In our context, the dominant language of multiplication is symbolic, though TT supports 192 a gestural form of expression as well as a visual one. Another aspect of the dominant lan-193 guage of multiplication is that of “groups of”, whose dominance will only diminish if we 194 make concerted efforts to translate multiplication into other languages” *Not clear what ‘our’ context is, could use some expansion, eg the TT app, or the French immersion, primary/secondary school/maths curriculum?

L.199 * Given the Diagne reference is in French and the focus on translation, just wonder if a brief (French original) Diagne quote + English translation might engage readers in some of the issues?   

l.203 “One way of respecting the “language” of each artefact-situation is to pay 204 attention not only to the words and gestures that are used, but also the material context, 205 which might include the ways of acting and moving that arise in the material environment 206 (which include the artefacts, the space, the people, etc.)” * It was not clear to me whether language is here being understood as an artefact/tool itself or an umbrella term for the whole artefact-situation, ie is the artefact-situation something which has a an identifiable ‘language’ (or languages?) of gestures, materials etc, or is the artefact-situation itself a language which can be translated into other languages? Cf “translation as not just involve words, 210 but visions of the world”

l.275 “We video-recorded these interviews to capture their discussion and their 275 actions while interacting with TT” * given French immersion school wondered what language participants were using (a) in their usual maths lessons (b) in observed play with TT (c) in interviews? In particular if any translations occurred as part of the study seems relevant to describe these?

l278 “The task consisted of two multiplicative word problems that relate to ideas of group-278 ing or unitising and array model, which corresponds to the two worlds more naturally.” * ‘(an) array model(s)’? If the material context is significant the context of the word problems/task could perhaps be described briefly – eg was it presented on paper/online etc?

l293 “In addition, students were given multiplicative statements in one world and were 294 explicitly asked to represent the same statement in the other world” * Could an example be included to help illustrate these statements, per the problems?

l303 “our languages are the different models of multiplication” * can this be squared with language as a vision of the world later on? i.e. is a model of multiplication a ‘world’, or is it say, reducable to a simple diagram of unary groupings vs array? If I understand the argument it may be that the act of translation itself complicates or simplifies these models.

l308 “When they move to 309 Zaplify, they do not experience it as another representation of multiplication; instead, it is 310 a new locality with its own ways of speaking and moving, but that also resonate with the 311 ones in Grasplify, even if only subconsciously. 312 “ * ‘resonate… subconsciously’ – this feels worth unpacking

l367 [pic] – suggest possibly rotating pics 180 degrees to make more readable? Here and later.

L425 “They then drew 426 horizontal and vertical lines between the pips and created additional dots where the lines 427 intersected (Fig 5a).” * wasn’t sure how drawing feature worked (sorry don’t have available touchscreen device to trial), is this literally drawing on the image?

l.436 *specify which app is pictured

l.444 * this pattern coding feels significant element of methodology maybe worth saying in methodology section that patterns would be coded

l513 “But when the interviewer 513 asked her to explain what she had done, she switched to Grasplify and said, “this is easier”” * Easier for her to solve the problem at the time, or potentially easier as a tool for explaining what she had done?

L609 “used strips of 608 paper to represent the hoodies and sweatpants and also did a one-to-one matching gesture 609 to pair each of the hoodies with one sweatpants.” *Interesting the choice to reduce the problem numerically and use strips of paper (ie another ‘artefact-situation’) – would be interested why the researchers ‘translated’ the problem in this way pedagogically at that juncture, and whether this led to translation from paper/simplified problem to TT

L665 “But they made 5 rows and 9 columns, even though the problem states that 666 there are 9 rows.” *Interesting how the connotations of rows change in translation of context – when ploughing/sowing perhaps the ‘row’ looks vertical from the plougher/sowers pov ie moving and working facing forward or looking backward along a furrow. And the 3D ‘up’ connotation of column as pillar etc. Perhaps students are still not familiar with the array-specific row/column terminology?

L727 “In our analysis, we were interested in how the 729 students translated across the world, wanting to pay attention to the various forms of ex-730 perience at play, including the actions, the interpretation of visual signs, the handling of 731 the quantities and the product, and the feelings associated with using each world (effi-732 ciency, ease of understanding, comfort, etc.)” *Are these the forms of experience that are ‘translated’? Are these models? Are these integral to the worlds or does the world exist outwith these? Or is it the shifts in these which are attended to?

L741 “as well as from the sequential appearance of the 741 factors in both written and oral forms of the problem statement.” * Translating from the 1D word problem experience or ‘world’ seems significant, is this explicitly a distinct world/translation, ie purely linear and linguistic, without 2D visuals?

Comments on the Quality of English Language

Overall very clear English, just a couple of minor typos picked up:

Abstract: “We study translation in terms of actions, strategies, perceptions and prefer-7 ences, highlights [highlighting?] both the translatable[s?] and the intranslatables that emerged in the pair-based inter-8 views that were conducted with grades 3-4 students.”

l.40 – * capitalise Cartesian

L.136 *typo coherent?

l.146 * consistency of artifact/artefact

l.149 * typo belongs?

Author Response

Many thanks for these outstanding suggestions. 

We have addressed both of the two suggestions. The issue of model versus representation is not one we wanted to delve in, but we used the language of the researchers we were citing and explained why we chose to use the term microworld. We also added more reflection on our use of Diagne and, specifically, on the different contexts in which Diagne is thinking.